# Polyphenol-Enriched Plum Extract Enhances Myotubule Formation and Anabolism while Attenuating Colon Cancer-induced Cellular Damage in C2C12 Cells

**DOI:** 10.3390/nu11051077

**Published:** 2019-05-15

**Authors:** Faten A. Alsolmei, Haiwen Li, Suzette L. Pereira, Padmavathy Krishnan, Paul W. Johns, Rafat A. Siddiqui

**Affiliations:** 1Food Chemistry and Nutrition Science Research Laboratory, Agricultural Research Station, College of Agriculture, Petersburg, VA 23806, USA; fatenalsolmei@gmail.com (F.A.A.); hali@vsu.edu (H.L.); 2Department of Biology, College of Natural and Health Sciences, Virginia State University, Petersburg, VA 23806, USA; 3Abbott-Nutrition Division, Research and Development, 3300 Stelzer Road, Columbus, OH 43219, USA; suzette.pereira@abbott.com (S.L.P.); paul.johns@abbott.com (P.W.J.); 4Valley Children’s Hospital, Madera, CA 93636, USA; padmakrishnan@hotmail.com

**Keywords:** cachexia, plum, cancer, muscle wasting, myoblasts, protein synthesis

## Abstract

Preventing muscle wasting in certain chronic diseases including cancer is an ongoing challenge. Studies have shown that polyphenols derived from fruits and vegetables shows promise in reducing muscle loss in cellular and animal models of muscle wasting. We hypothesized that polyphenols derived from plums (*Prunus domestica*) could have anabolic and anti-catabolic benefits on skeletal muscle. The effects of a polyphenol-enriched plum extract (PE60) were evaluated in vitro on C2C12 and Colon-26 cancer cells. Data were analyzed using a one-way ANOVA and we found that treatment of myocytes with plum extract increased the cell size by ~3-fold (*p* < 0.05) and stimulated myoblast differentiation by ~2-fold (*p* < 0.05). Plum extract induced total protein synthesis by ~50% (*p* < 0.05), reduced serum deprivation-induced total protein degradation by ~30% (*p* < 0.05), and increased expression of Insulin-Like Growth Factor-1 (IGF-1) by ~2-fold (*p* < 0.05). Plum extract also reduced tumor necrosis factor α (TNFα)-induced nuclear factor κB (NFκB) activation by 80% (*p* < 0.05) in A549/NF-κB-luc cells. In addition, plum extract inhibited the growth of Colon-26 cancer cells, and attenuated cytotoxicity in C2C12 myoblasts induced by soluble factors released from Colon-26 cells. In conclusion, our data suggests that plum extract may have pluripotent health benefits on muscle, due to its demonstrated ability to promote myogenesis, stimulate muscle protein synthesis, and inhibit protein degradation. It also appears to protect muscle cell from tumor-induced cytotoxicity.

## 1. Introduction

Skeletal muscle weakness and wasting, which is also referred as cachexia, is a major clinical problem for advanced cancer patients [1]. In 1932, Warren described cachexia as the most common cause of death across a variety of cancers in a post mortem study of 500 patients [2]. The term “Cachexia” is derived from the Greek words “kakos” and “hexis,” meaning “bad condition.” It is a multi-organ syndrome associated with and characterized by at least 5% body weight loss due to muscle and adipose tissue wasting [3]. Cancer cachexia is a multifactorial syndrome that is common in advanced malignancy occurring in 80% of patients, which cannot be reversed by nutritional support and leads to significant function deficits [4], and which is responsible for an estimated 20% of cancer-related deaths [5].

Colorectal cancer (CRC) patients are often presented with cachexia syndrome, which is a major contributor to colorectal cancer-related morbidity and mortality [4,5,6,7,8]. About 35 to 60% of CRC patients show some degree of muscle wasting and 28% lose >5% of their body weight in the six months preceding diagnosis [9]. Blocking muscle wasting can prolong life even in the absence of effects on tumor growth [10]. 

Oxidative stress through activating initial steps in protein degradation via the ubiquitin-proteasome pathway and the activation of caspases contributes to muscular atrophy [11,12,13]. In addition, inflammation also leads to muscle atrophy and this is mediated through cytokine (e.g., tumor necrosis factor α (TNFα), interleukin-6 (IL-6), and interferon γ (IFNγ)) induced activation of the nuclear factor κB (NF-κB) pathway [14].

Recent studies have shown that polyphenol-rich plant extracts prevent oxidative stress, reduce inflammation, and help reduce muscle atrophy. We have previously shown that curcumin treatment attenuated muscle wasting in cancer cachectic mice [15]. Supplementation with red grape polyphenols mitigated muscular atrophy in transforming growth factor (TGF) mice, a model of chronic inflammation, by reducing mitochondrial oxidative stress and by inhibiting caspase activation [16]. Grape seed extract supplementation effectively prevented muscle wasting in IL10-knock out mice [17]. Green tea polyphenol, catechins, protected normal and dystrophic muscle cells from oxidative damage [18]. Epigallocatechin-3-gallate (EGCG) supplementation preserved muscle in sarcopenic rats [19] and attenuated skeletal muscle atrophy caused by experimentally induced cancer cachexia in in mice [20]. More recently, ursolic acid—a polyphenol present in apple peels, basil leaves, prunes, and cranberries [21]—has been shown to increase muscle mass in mice exhibiting fasting-induced muscle atrophy [22]; it has also increased muscle mass, fast and slow fiber size, grip strength, and exercise capacity in mice with diet-induced obesity [23]. These observations clearly suggest that intake of polyphenols can be beneficial in preserving muscle mass.

The common plum (*Prunus domestica*) is well known to be rich in polyphenols and contains unique phytonutrients called neochlorogenic and chlorogenic acid which have high antioxidant activities. Among functional foods, plums are also considered “super foods” since their consumption has been associated with the decrease in chronic degenerative diseases and circulatory and digestive issues [24]. Dried plums have been shown to reduce symptoms of arthritis in an inflammation model [25]. These effects are attributed to their high polyphenolic composition and related high antioxidant activity [26]. Plums have several health benefits and studies have found that plums also initiate anti-cancer mechanisms that may help prevent the growth of cancerous cells and tumors [27,28,29]. 

In addition, plums have been extensively studied for their effects on bone health [30,31]. Plums contain caffeic acid (the polyphenol component of neochlorogenic and chlorogenic acids) and rutin, which have been shown to inhibit the deterioration of bone tissues and prevent diseases such as osteoporosis in postmenopausal women [32]. Research has also shown that regular consumption of dried plums helps in the restoration of bone density lost to aging [33].

Formation of bone and much of the skeletal tissues is derived from the proliferation and differentiation of skeletal stem cells. As dried plum was found to be a potent regulator of bone health, it is possible that plum and its associated polyphenols may have benefits on other cells of musculoskeletal system. Thus, in the present study, we sought to investigate the effect of a polyphenol-enriched plum extract on muscle cell growth and differentiation, and on muscle protein synthesis and degradation *in vitro*. In addition, we explored the effect of plum extract on inflammation as well as studied its effect on colon cancer cells. 

## 2. Materials and Methods 

### 2.1. Materials

Dulbecco’s modified Eagle’s medium (DMEM), fetal bovine serum (FBS), horse serum, and Penicillin-Streptomycin solution were purchased from Gibco-Thermo-Fisher Scientific (Grand Island, NY, USA). L-[2,3,4,5,6-3H] Phenylalanine and L-[Ring-3, 5-3H]-Tyrosine was purchased from Perkin-Elmer (Waltham, MA, USA), Prune extract-60% enriched polyphenol extract (PE60) was purchased from PL Thomas (Morristown, NJ, USA). All other chemicals were of reagent grade, and were purchased from Sigma Chemical Co. (St. Louis, MO, USA). 

### 2.2. Composition of the PE60-Plum Extract

Free gallic acid, 3-cholorogenic acid, rutin, free quercetin, and proanthocyanidins were determined with an Agilent Technologies (Wilmington, DE, USA) Model 1200 HPLC System equipped with a Model G1311A quaternary pump, Model G1322A vacuum degasser, Model G1329A autosampler, Model G1316A thermostatted column compartment, a Model G1315B diode array detector, and a Chem Station data processor. The separations were performed with a YMC-Pack ODS-AQ analytical column (4.6 × 250 mm, 5 µm, P/N AQ12S05-2546WT, Waters Corporation, Milford, MA, USA), using mobile phase A = 1000/100 (*v*/*v*) 0.05 M KH2PO4, pH 2.9/acetonitrile, and mobile phase B = 200/800 (*v*/*v*) Milli-Q Plus water/acetonitrile, a column temperature of 40 °C, an injection volume of 5 µL, and the analytes were quantified at signals of 280 nm/590 nm (for gallic acid and the proanthocyanidins), 330 nm/590 nm (for 3-chlorogenic acid), and 375 nm/590 nm (for rutin and quercetin). The elution program was 0% mobile phase B from 0 to 5 min, 0 to 60% (linear gradient) mobile phase B from 5 to 35 min, 100% mobile phase B from 35 to 40 min, and 0% mobile phase B from 40 to 55 min (end). The PE60 extract was prepared for analysis by stirring (at room temperature for 15 min) 0.250 g in 100 mL of 50/50 (*v*/*v*) 0.05 M citric acid/methanol. The determinations were calibrated with standard solutions of gallic acid, 3-chlorogenic acid, rutin hydrate, and quercetin dihydrate (all obtained from Sigma-Aldrich, St. Louis, MO, USA), also prepared in the citric acid/methanol medium. The proanthocyanidin content was estimated by peak area proportionation vs. the corresponding peak areas (at 280 nm/590 nm) of grapeseed extracts (from Kikkoman, Polyphenolics, and Seppic) of known (i.e., label claim) proanthocyanidin content, included in the analysis. The anthocyanin concentration was estimated by a published colorimetric method [34]. During present investigation, minor isomers of chlorogenic acid (4-chlorogenic acid, 5-chlorogenic acid) were not determined. 

### 2.3. Characterization of Anti-Oxidation Capacity of the Plum Extract

The PE60 (Lot PE6009-1601) extract was dissolved in water (10 mg/mL) and then centrifuged at 1500 × *g* for 10 min to remove any insoluble material. The dissolved material was sterile filtered and the filtrate was assayed for total polyphenols by the Folin Ciocalteu method [35], for total flavonoids by the AlCl3 complexation method [36], for anti-oxidant activity by the DPPH assay [37], and for oxygen scavenging activity by the ABTS assay [38], as described. 

### 2.4. Cell Culture

C2C12 cell line (mouse myoblasts) were obtained from American Type Culture Collection (Manassas, VA, USA). The undifferentiated cells were grown in complete media consisting of Dulbecco’s modified Eagle’s medium (DMEM, 4.5 mg/mL glucose) supplemented with heat-inactivated fetal calf serum (10%), penicillin (100 units/mL), and streptomycin (100 μg/mL) at 37 °C in the presence of 5% CO_2_. The myoblasts were differentiated into myotubes by culturing them into differentiation medium, consisting of DMEM supplemented with heat-inactivated horse serum (5%), penicillin (100 units/mL), and streptomycin (100 μg/mL) for five days. 

### 2.5. Determination of C2C12 Myoblast Cell Size

Muscles cells were grown in a 96-well plate for 24 h in 100 μL complete media. Cells were then treated with 0, 50, 100, 150, and 200 μg/mL of extract for 48 h to evaluate a dose-response effect of plum extract. After incubations, the cells were observed under a microscope and pictures (100 × magnification) were taken using a Nikon microscope with calibrated objectives. The size of cells was determined using Element-BR software (Nikon Instruments Inc, Melville, NY, USA). 

### 2.6. Assaying C2C12 Myoblast Differentiation

Muscle cells were initially cultured in a 96-well plate for 24 h in 100 μL complete media. Cells were then incubated with 0, 50, 100, and 200 μg/mL plum extract for five days and the medium containing corresponding concentration of plum extract was changed every 24 h. After treatment, the cells were washed once with PBS, and then fixed with cold 4% paraformaldehyde for 10 min on ice. The cells were washed three times with PBS and the monolayer was treated with blocking solution containing 2% albumin. The cells were then incubated with anti-myosin antibody at room temperature for 2 h. Cell were washed again and then incubated with anti-mouse Alexa-488 antibody (Abcam, Cambridge, MA) for two hours. Cells were washed again three times with PBS and the nuclei were stained briefly with Hoechst 33342 dye (1:2000 dilution). Pictures were taken at 200 × magnification using a Nikon Fluorescent Microscope (Nikon Instruments Inc, Melville, NY 11747, USA). Myotubes were defined as myosin positive cells with 2 or more fused nuclei. 

### 2.7. Protein Synthesis in Cultured C2C12 Myotubules

C2C12 cells (375,000) were initially plated on a 12-well tissue culture plate that was initially coated with 2% gelatin. Cells were differentiated for five days in 5% horse serum (media was changed every two days) and then starved for 30 min by replacing the media with 1 ml PBS. The cells were then treated with 0, 50, 100, and 200 μg/mL of plum extract in PBS, spiked with [^3^H] phenylalanine (1μCi/well), and incubated for 2 h at 37 °C. The reaction was stopped by placing the plates on ice. Wells were washed two times with DPBS-media containing 2 mM cold phenylalanine. Further, 1 mL of 20% cold trichloroacetic acid (TCA) solution was added to each well and plates were incubated on ice for 1 h for protein precipitation. Wells were washed two times with cold TCA and then the precipitated proteins were dissolved in 0.5 mL of 0.5N NaOH containing 0.2% Triton X-100 overnight in a refrigerator. An aliquot (5 μL) of the NaOH solubilized material was used for protein determination and the rest of the dissolved proteins were mixed with scintillation fluid and counted. Data is computed as cpm/mg of proteins and then % change over control is calculated. 

### 2.8. Protein Degradation in C1C12 Myotubules

C2C12 myoblasts were cultured and differentiated as described above. Cells were then labelled with [^3^H] Tyrosine 1 μCi/1 mL in serum free-DMEM (SF-DMEM) for 24 h. The unincorporated [^3^H] Tyrosine was removed by washing the cell monolayer three times with SF-DMEM containing 50 μM cycloheximide (protein synthesis inhibitor) and 2 mM non-labelled Tyrosine. Proteolysis was induced by serum deprivation for 48 h in the presence or absence of 50, 100, 200 μg/mL of plum extract in serum-free DMEM containing 50 μM cycloheximide. The extent of protein degradation was assayed by monitoring release of radioactive tyrosine in the media after 48 h of incubation and was expressed as protein degradation in comparison to control (normalized to 100%). 

### 2.9. Determination of Insulin-Like Growth Factor-1 (IGF-1) Expression

Total RNA was extracted from C2C12 myotubules with RNeasy Plus Universal Mini Kit (Qiagen, Hilden, Germany), according to the manufacturer’s instructions. The concentration and purity of RNA was determined by measuring the absorbance in a Nano drop spectrophotometer. RT2 First Strand Kit from Qiagen (Qiagen, Hilden, Germany) was used to synthesize first strand complementary DNA (cDNA). The gene expression levels were analyzed by Quantitative real-time RT-PCR conducted on the Bio-Rad CFX-96 Real-Time PCR System using RT2 SYBR Green Master mix (Bio-Rad Laboratories, Hercules, CA). The primers (*IGF*: forward primer GGACCAGAGACCCTTTGCGGGG and reverse primer, AGCTCAGTAACAGTCCGCCTAGA; *GAPDH*: forward primer ATCCCATCACCATCTTCCAG and reverse primer CCATCACGCCACAGTTTCC) were designed. Hot-Start DNA Taq Polymerase was activated by heating at 95 °C for 10 min and real time PCR was conducted for 40 cycles (15 s for 95 °C, 1 min for 60 °C). All results were obtained from at least three independent biological repeats. Data were analyzed using the ΔΔCT method. Glyceraldehyde-3-phosphate dehydrogenase (*GAPDH*) genes were used as house-keeping genes for expression calculation. 

### 2.10. Determination of NFkB Activation

A549/NFkB-luc cells (Panomics Catalog No. RC0002) at 3 × 10^5^/well were seeded in 1 mL of Initial Growth Media (Dulbecco’s Modified Eagle’s medium containing 10% FBS and 1% Pen-Strep) in a 12-well plate. The cells were incubated in a humidified incubator at 37 °C and 5% CO_2_ for 24 h to allow cells to recover and attach. After washing the cells once with serum-free media containing penicillin (100 units/mL), and streptomycin (100 mg/mL), 1 mL of this media was added to each well. Cells were pretreated with varying concentrations of plum extracts for 1 h at 37 °C and 5% CO_2_, and then TNFα was added to achieve a final concentration of 2 ng/mL to all wells except control untreated cells. The cells were incubated in a humidified incubator at 37 °C and 5% CO_2_ for 6 h. After treatment, the media was carefully removed. Cells were washed with PBS once and then lysed by 100 μL of 1× lysis buffer. Assay for luciferase activity was performed according to assay manufacturer’s (Promega P/N E1500) recommendations. The average relative luminescence units (RLU) were calculated and corrected for baseline quenching for each set of triplicate wells, using WinGlow software (PerkinElmer, Waltham, MA 02451, USA and Microsoft Excel (Microsoft Corporation, Redmond, WA 98073, USA). The data is reported as the relative percent inhibition of TNFα mediated NFκB activation on A549 cells.

### 2.11. Effect of Plum Extract on Colon-26 Proliferation and its’ Soluble Factor Induced Cytotoxicity on C2C12 Myotubules

Colon 26 cells, a mouse colon carcinoma cell line, was obtained from American Type Culture Collection (Manassas, VA, USA). Effect of plum extract on Colon-26 cell proliferation was assayed using a Water-Soluble Tetrazolium-1 (WST-1) (Talkara, Shiga, Japan) assay as described previously [39]. To determine the effects of soluble factors released from Colon-26 on C2C12 myotubules, conditioned media from Colon-26 culture was collected after 24 h. of cultivation. The media was centrifuged at 2500 × *g* for 20 min to remove cellular material. The clear supernatant (conditioned media) was diluted 1:10 with normal complete media. The C2C12 differentiated myoblasts were then treated with normal complete medium or with Colon-26 conditioned medium with or without 50 μg/mL plum extract. A lower dose of plum extract (50 μg/mL) was used to avoid a direct effect of higher dose of plum extract (100 μg/mL or 200 μg/mL) on protein synthesis and degradation. The cell viability was assayed using a WST-1 assay. Control cells were subjected to equal amounts of non-conditioned media.

### 2.12. Data Analysis

The data is expressed as mean ± SD for at least three replicates. All comparisons were made by one-way ANOVA with Tukey’s -HSD-post-hoc test using SPSS Statistics 20 software. All significant differences are reported at *p* < 0.05 and indicated by “*”.

## 3. Results

### 3.1. Characterization of PE60 Plum Extract Composition and Anti-Oxidation Properties

As shown in Table 1, the major components identified in the polyphenol-enriched PE60 plum extract are proanthocyanidins, along with minor components such as anthocyanidins, 3-chlorogenic acid, rutin, quercetin (free), and gallic acid (free). 

The PE60 was also characterized by determining total phenolic content (TPC), total flavonoid content (TFC), anti-oxidant activity (DPPH assay), and oxygen scavenging activity (ABTS). The data in Table 2 shows that the content of TPC was in the same range as reported by the commercial vendor (60%). The data indicate that the PE60 contained TPC in range 525–575 mg/g of dry extract. The TFC was in 480–560 mg/g dry weight range. The anti-oxidation effects as determined by inhibition of DPPH oxidation and ABTS assay ranged from 3280–3460 and 4000–4500 μM Trolox equivalents/g, respectively. 

### 3.2. Effect of PE60 Plum Extract on C2C12 Myoblast Size and Differentiation

Plum extract had no cytotoxic effect on myoblast when used even at a high dose of 250 μg/mL (data not shown). It is evident from images that plum extract has some effect on cell proliferation; however, it was interesting to note that the plum extract increased the size of undifferentiated myoblasts cells in a dose-dependent manner (Figure 1a). The size of myoblast increased ~two-fold (*p* < 0.05) after treating cells with 50 μg/mL of plum extract when compared to that of untreated-control cells. Increase in myoblast size plateaued to a maximum increase of three-fold at 200 μg/mL concentration (Figure 1b). The effect of plum extract was also assessed on myoblast differentiation. Figure 2a indicates that the plum extract stimulated differentiation of myoblast in a dose-dependent manner using expression of myosin heavy chain as a marker for differentiation. The number of myotubes formed resulting from fusion of differentiated cells was increased by two-fold in cells treated with 100 μg/mL plum extract (*p* < 0.05) and by three-fold at 200 μg/mL (*p* < 0.05) compared to that of control cells (Figure 2b). 

### 3.3. Effect of PE60 Plum Extract on Myotubule Protein Synthesis

Plum extract showed almost a linear increase in [^3^H] phenylalanine incorporation into proteins in a dose dependent manner in C2C12 myotubules (Figure 3). Doses of 100 μg/mL and 200 μg/mL of plum extract caused a significant increase in protein synthesis by 30% and 50%, respectively (*p* < 0.05).

### 3.4. Effect of PE60 Plum Extract on Myotubules Protein Degradation

We also examined if plum extract could reduce myotubule protein degradation induced by serum starvation. Figure 4 revealed that plum extract did inhibit protein degradation in a dose-dependent manner. Doses of 100 μg/mL and 200 μg/mL significantly inhibited protein degradation by 20% and 30%, respectively (*p* < 0.05). 

### 3.5. Effect of PE60 Plum Extract on IGF-1 Expression in Myotubules

Expression of IGF-1 mRNA in C2C12 myotubules upon treatment with plum extract is shown in Figure 5. Compared to that of untreated cells, low concentration of plum extract (50 μg/mL) has no significant effect on IGF-1 mRNA expression; however, it significantly stimulated IGF-1 expression when cells were treated at a higher dose (100 or 200 μg/mL) plum extract. 

### 3.6. Anti-Inflammatory Effect of PE60 plum Extract in Vitro

We evaluated the anti-inflammatory activity of plum extract by assessing its effect on TNF-α-induced NFkB activation where the activity was measured in terms of luciferase activity of NFkB reporter system assay. Plum extract inhibited NFkB activation in a dose dependent manner (Figure 6). A dose response assay indicated that ~40% inhibition (*p* < 0.05) of TNF-α-mediated NFkB activation was achieved at 25 μg/mL plum extract, and >80% inhibition (*p* < 0.05) of TNF-α-mediated NFkB activation was achieved at 50 μg/mL plum extract. 

### 3.7. Effect of PE60 Plum Extract on Colon-26 Mouse Adenocarcinoma Cell Line 

When Colon-26 cells were treated with plum extract, the cells viability was reduced in a dose-dependent manner reaching ~80% reduction (*p* < 0.05) at 150 μg/mL. Upon further increasing the concentration of plum extract, the cell viability was further reduced 90% (*p* < 0.05) at 200 μg/mL (Figure 7). 

### 3.8. Effect of PE60 Plum Extract on C2C12 Cell Viability in Response to Colon-26 Cells Cytotoxicity-Inducing Factors

Mouse derived Colon-26 adenomacarcinoma cells are known to induce muscle wasting in rodents [40]. The effect of these circulating soluble factors released by Colon-26 was examined on growth of C2C12 myotubules in vitro in presence or absence of plum extract. Figure 8a,b shows that in the absence of plum extract, soluble factors released in media derived from Colon-26 cells caused a significant reduction of C2C12 cell viability by ~25% (*p* < 0.05). However, in the presence of plum extract, the negative effects of Colon-26-derived media on C2C12 viability was prevented and the cell viability was maintained to a similar level that was seen in the untreated cells.

## 4. Discussion

In our study, we sought to investigate if plums had benefits on skeletal muscle. Specifically, we selected to use a plum extract that was enriched in polyphenols (~60% polyphenols) because the health benefits of plum have been partly attributed to its high polyphenol content [41,42,43]. Our data indicates that about 95% of total phenolic content in the plum extract used was present in the form of flavonoids. This data is not surprising as fruits are often reported to have phenolic compounds which are high in flavonoids with a range of 90–100% [44]. The anti-oxidant activity in the plum extract was found to be in range of 3–4 mM of Trolox equivalent/g, which is higher than that of turmeric (0.27–0.35 mM Trolox eq/g) and mulberry (1–2 mM Tolox eq/g), but lower than green tea (13–17 mM Trolox eq/g) and pomegranate (20–25 mM Trolox eq/g) [45,46,47,48,49]. 

Dried plum has previously been reported to have health benefits on bone. In rat models of osteoporosis, dried plum intake resulted in prevention and reversal of bone loss [50,51]. A three-month clinical intervention study showed that dried plum intake improved biomarkers of bone formation in postmenopausal women, whereas longer-term intake of dried plum resulted in mitigating loss of bone mineral density [31]. The present study was designed to analyze the effects of plum extract on muscle metabolism in C2C12 myotubules. In our initial experiments, the effect of plum extract was tested on myoblast viability. The data show that this plum extract has no toxicity on the muscle cells, even at very high doses. These results are consistent with prior literature on plum effects on non-diseased cells [52]. The maintenance of muscle mass is dependent on synthesis of new proteins and breakdown of old or damaged proteins. If these processes are balanced, the muscle mass is maintained; however, with aging and under certain catabolic condition including cancer, renal failure or trauma, muscle protein degradation exceeds the synthesis of new proteins, and results in muscle atrophy [53]. One interesting observation was that that plum extract increased the size of growing myoblast under un-differentiated conditions, suggestive of inducing increase in cytoplasmic volumes by stimulating protein synthesis. We also measured effect of plum extract on protein synthesis and degradation in differentiated myotubules. Our data clearly demonstrated that plum extract not only increased protein synthesis but also inhibited myotubules protein degradation in response to serum starvation, demonstrating both an anabolic and anti-catabolic effect. 

The activity of the plum extract appears to be at least partly mediated through IGF-1 stimulation. Several studies have shown that IGFs stimulated both proliferation and differentiation of myoblasts, and also play a role in regenerating damaged skeletal muscle [54,55,56,57,58]. In line with our results, prior studies have also demonstrated that plums can increase IGF-1 levels in both humans [58] and animal models [51,59]. One of the manifestations of muscle loss is associated with decreased production of IGF-1 [60]. The signaling pathway IGF-1/PI3K/Akt (Insulin like growth factor -1/phosphatidyl inositol 3-kinase/protein kinase) is considered the main mediator of normal muscle development and one of the most studied signaling molecular systems involved in muscle metabolism [61]. Akt activation leads to activation of mTOR (mammalian target of rapamycin), which is responsible for promoting protein synthesis. The Akt-mTOR signaling pathway and its downstream components (p70s6k and 4E-BPI) are attenuated with muscle wasting [62]. Further studies need to be performed to confirm the if plum extract is indeed regulating Akt activity. The identification of compound or compounds in plum responsible for stimulating IGF-1 levels in myoblast was beyond the scope of the present study. As discussed earlier, ursolic acid has been shown to increase muscle mass in mice exhibiting fasting-induced muscle atrophy [22] or diet-induced obesity [23]. Interestingly, ursolic acid has also been shown to induce IGF-1 levels in the skeletal muscle of these mice with an increased Akt phosphorylation [22,23]. During present investigation, we were not able to detetect ursolic acid in PE60 extracts due to technical limitation for detecting all polyphenols; however, other studies have reported presence of ursolic acid in plums [21]. Therefore, it is possible that ursolic to some extent may have contributed in IGF-1 mediated muscle growth in our studies. 

Studies have demonstrated the anti-inflammatory effect of dried-plum or plum juice in several cellular system including lipopolysaccharide-induced macrophages [63,64], splenocytes from ovariectomized mice [65], colorectal cells in azoxymethane-treated rats [66], heart tissues in obese rats [67] and joints of TNF-over expressing mice [25]. The antioxidation activities of plum appeared to be mediated through the inhibition of NFκB activation [25,66,67]. Based on these reported studies, we decided to test the effect of plum extract on NFκB activation, since oxidative stress and inflammatory responses through activation of NFκB play an important part in muscle atrophy. Activation of NFkB plays a central role in muscle atrophy in several catabolic situations including cancer cachexia [68,69]. We found that even a small dose of plum extract was able to almost completely inhibit (>80% inhibition) TNF-α-induced NFκB activity *in vitro*. It is likely that the proanthocyanidins, which comprise over 70% of the polyphenols, may be involved in suppressing the inflammatory cytokine (TNF)-induced activation of NFκB, although this has not been systematically tested with the individual components of the extract.

Cancer cachexia-related morbidity and mortality are often accompanied by whole body and muscle loss [4,7,8] and it is suggested that blocking muscle wasting can prolong life despite tumor growth [10]. The effect of plum extract on colon cancer cell viability, as well as its ability to protect muscle cells from colon-cancer cell induced cytotoxicity, were, therefore, also investigated. We used Colon-26 adenocarcinoma cells, which is a widely used preclinical model because it induces clinical cachexia, including its development as well as the resultant physiological and metabolic impairment [40,69,70]. Treating the Colon-26 colon cancer cells with plum extract caused a significant decrease in the Colon-26 cell’s viability, indicating potential anti-tumor activity. 

It is known that muscle wasting in cancer patients is mediated through factors released from tumor in circulation [71,72,73,74]. Studies have shown that elevated circulating levels of IL-6 mediated skeletal muscle cell death in severely cachectic mice with colon cancer [75]. Our studies found that plum extract can protect C2C12 myotubules from cytotoxicity induced by soluble factors released by the Colon-26 cells. The exact pathways leading to reduced cell viability in response to tumor induced soluble factors are not known, but it is possible that both atrophy and apoptosis may be attenuated by the plum extract. It is also possible that compound(s) in plum extract may directly affect colon cells to inhibit secretion of inflammatory cytokines. Future studies need to be conducted to elucidate the molecular mechanism involved in the anti-cytotoxic activity of the plum extract.

Our current studies have several limitations. The study was performed using an *in vitro* system that may not represent the complexities of an *in vivo* system. Furthermore, polyphenols in the plum extract can undergo biotransformation in vivo, which could either enhance or diminish the anabolic of plum extract on muscle as well as its anti-inflammatory benefit. However, previous human studies with dried plum still demonstrated its ability to activate IGF-1 as well as its anti-inflammatory benefits, indicating that biotransformation may not result in loss of these effects observed in our study.

## 5. Conclusions

In conclusion, the polyphenol-enriched plum extract has both anti-catabolic and anabolic effects on muscle cells, as well as myogenic potential. In addition, this plum extract exhibited anti-cytotoxic properties in response to soluble factors released from cancer cells. Thus, plum extract may be a useful intervention to be considered for cancer cachexia or other chronic disease-induced cachexia involving inflammation. These results need to be confirmed in an animal model of cachexia, followed by clinical translation. 

## Figures and Tables

**Figure 1 nutrients-11-01077-f001:**
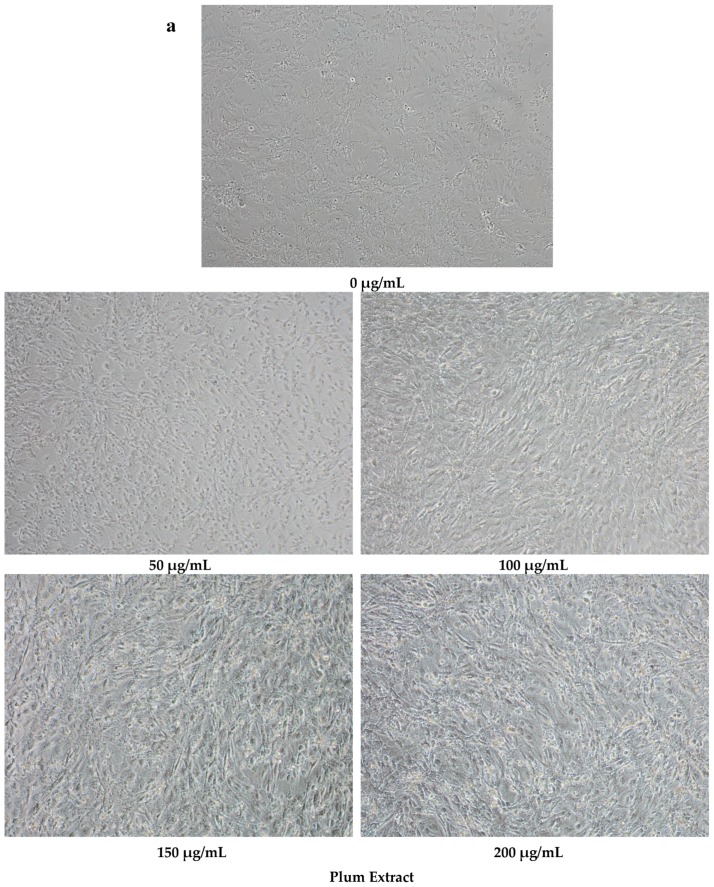
The effect of plum extract on C2C12 myoblast cell size. (**a**) The representative pictures of myoblast after treatment with varying concentration of plum extract (100 × magnified images) taken by a Nikon Microscope. The bar represents a length of 500 μm. (**b**) The size of myoblast was determined using Element-BR software as described in “Materials and Methods”. The data are expressed as mean ± SD for at least three experiments. All comparisons were made to control (untreated cells) using one-way ANOVA; significant differences are reported at * *p* < 0.05.

**Figure 2 nutrients-11-01077-f002:**
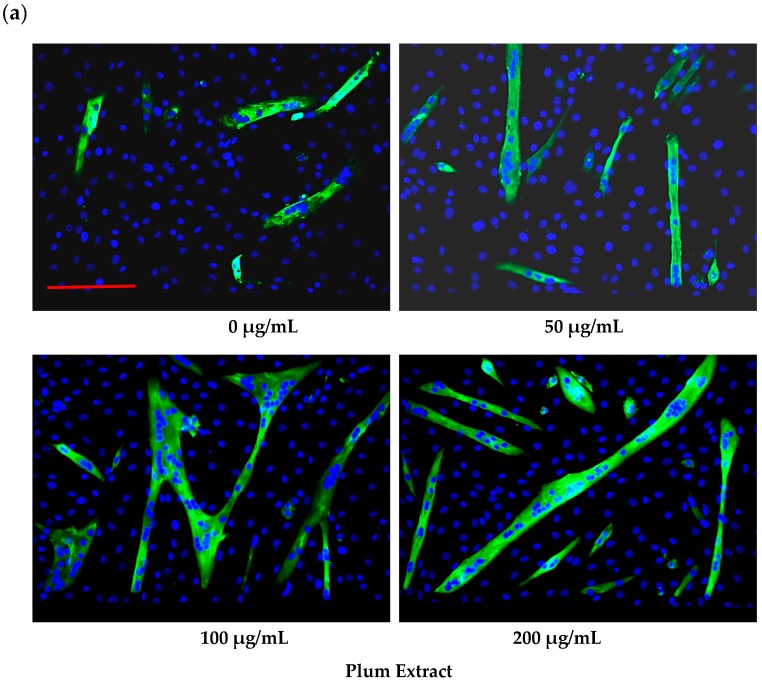
The effect of plum extract on C2C12 myoblast differentiation. (**a**) Images of differentiated cells after treatment with varying concentration of plum extract showing nuclei stained in blue (Hoechst 33342) and myofibers stained in green (Alexa 488). Pictures were taken at 200× magnification using a Nikon Fluorescent Microscope. The bar represents a length of 300 μm. (**b**) Fused cells from five random fields were counted manually under 200× as described in “Materials and Methods”. The data are expressed as mean ± SD for at least three experiments. All comparisons were made to control (untreated cells) using one-way ANOVA; significant differences are reported at * *p* < 0.05.

**Figure 3 nutrients-11-01077-f003:**
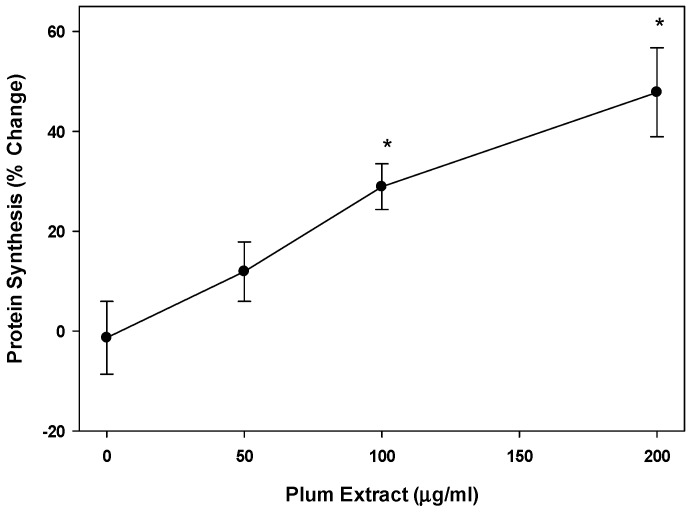
The effect of plum extract on myotubule protein synthesis. Protein synthesis was measure by the incorporation of labeled phenylalanine into total myotubule proteins in response to various levels of plum extract. Data were computed as cpm/mg of proteins followed by calculation of % change over control. The data were expressed as mean ± SD for at least three experiments. All comparisons were made to control (untreated cells) using one-way ANOVA; significant differences are reported at * *p* < 0.05.

**Figure 4 nutrients-11-01077-f004:**
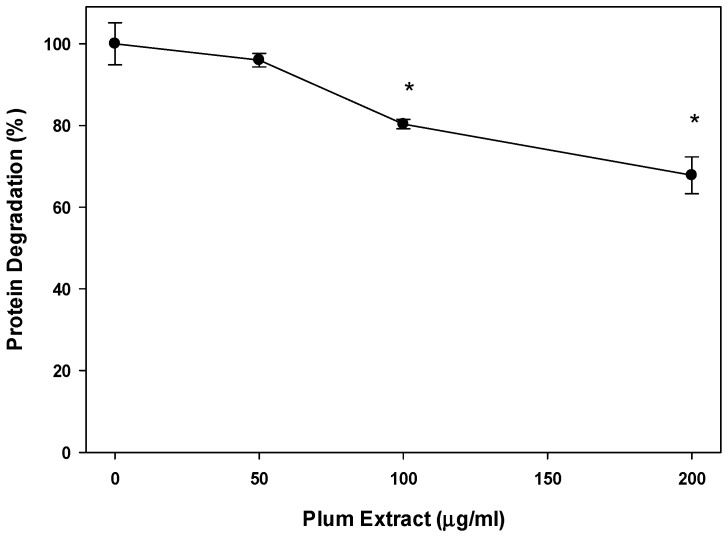
The effect of plum extract on myotubule protein degradation. Proteolysis was induced by 48 h-serum starvation in the presence or absence of plum extract, and monitored by release of radioactive tyrosine from pre-labelled cells. Data were computed as cpm/mg of proteins and then % change over control was calculated. The data were expressed as mean ± SD for at least three experiments. All comparisons were made to control (untreated cells) using one-way ANOVA; significant differences are reported at * *p* < 0.05.

**Figure 5 nutrients-11-01077-f005:**
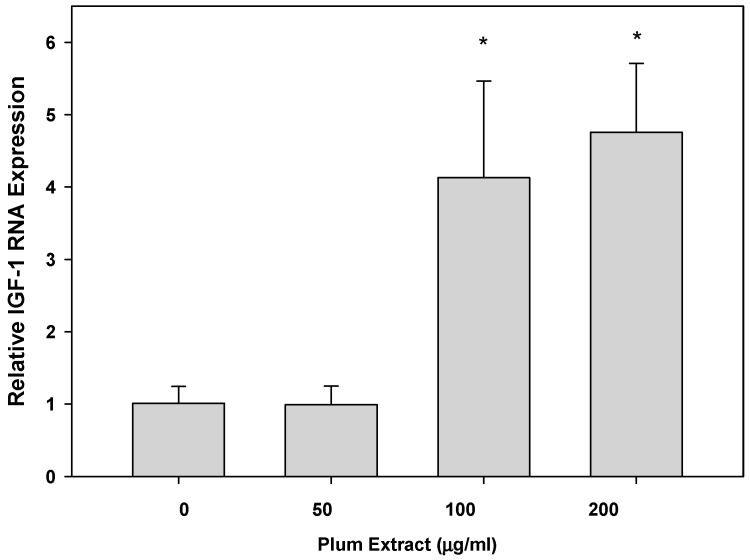
The effect of plum extract of IGF-1 gene expression. Total RNA was extracted from C2C12 myotubules treated with various concentrations of plum extract and compared to untreated control. All results were obtained from at least three independent biological repeats. Data were analyzed using the ΔΔCT method. Glyceraldehyde-3-phosphate dehydrogenase (*GAPDH*) genes were used as house-keeping genes for expression calculation. All comparisons were made to control (untreated cells) using one-way ANOVA; significant differences are reported at * *p* < 0.05.

**Figure 6 nutrients-11-01077-f006:**
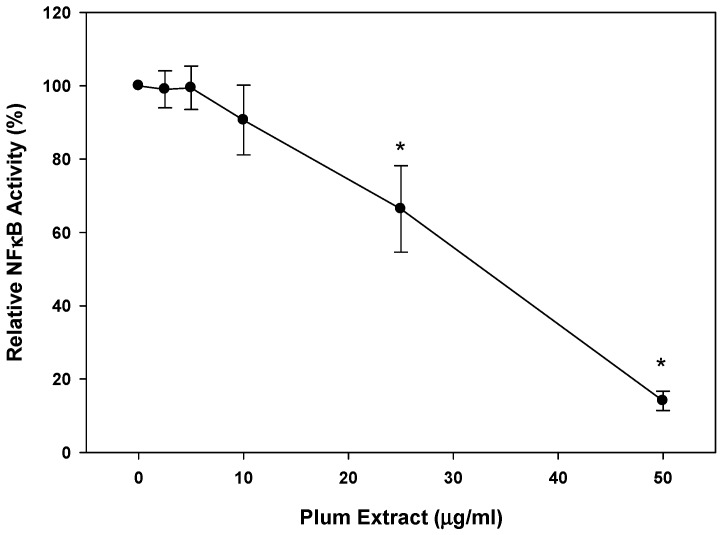
Effects of plum extract on NFkB activation. The effect of plum extract on TNFα-mediated NFkB activation was measured in the A549/NFκB-luc reporter stable cell line. Activity was measured in terms of luciferase activity. The data are reported as the relative percent inhibition of TNFα-mediated NFkB activation. The data are expressed as mean ± SD for at least three experiments. All comparisons were made to control (untreated cells) using one-way ANOVA; significant differences are reported at * *p* < 0.05.

**Figure 7 nutrients-11-01077-f007:**
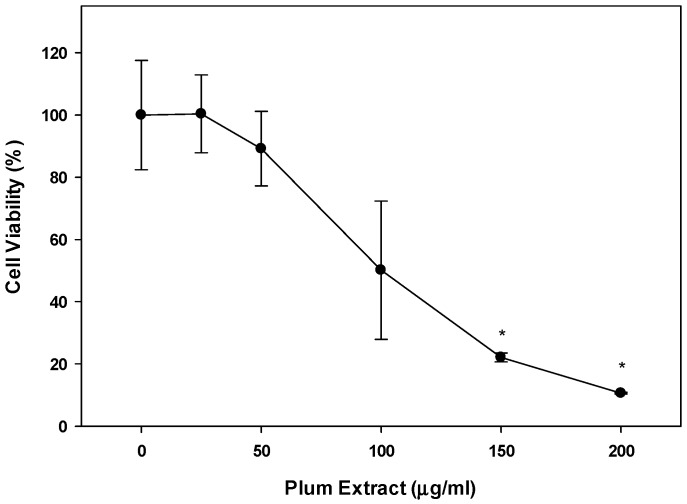
Effect of plum extract on Colon-26 adenocarcinoma cells. Data were calculated as % inhibition of cell growth in response to various concentrations of plum extract. The data are expressed as mean ± SD for at least three replicates. All comparisons were made to control (untreated cells) using one-way ANOVA; the significant differences are reported at * *p* < 0.05.

**Figure 8 nutrients-11-01077-f008:**
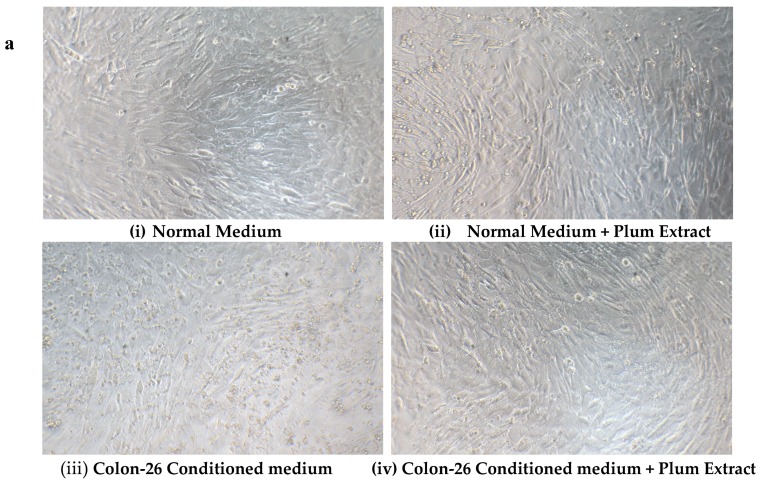
The effect of plum extract on C2C12 viability in response to Colon-26-induced cytotoxicity. (**a**) Differentiated C2C12 myotubes were treated with normal medium (i & ii) or Colon-26-conditioned medium (iii & iv) in the absence (i & iii) or presence (ii & iv) of plum extract (50 μg/mL). (**b**) The viability of C2C12 myotubules were determined using WST-1 assay. The data is expressed as mean ± SD for at least three experiments. All comparisons were made to control (untreated cells) using one-way ANOVA; significant differences are reported at * *p* < 0.05.

**Table 1 nutrients-11-01077-t001:** Characterization of composition of polyphenol-enriched plum extract (PE60).

Component	Concentration (g per 100 g) *n* = 3	Flavonoid Type (USDA)	Analytical Method
Anthocyanins	0.391 ± 0.020 (rsd = 5.1%)	Anthocyanidin	Colorimetric
3-chlorogenic acid	1.76 ± 0.01 (rsd = 0.6%)	Hydroxycinnamic acid	LC/UV *
Rutin	1.12 ± 0.01 (rsd = 0.6%)	Flavanol	LC/UV *
Quercetin (free)	0.718 ± 0.005 (rsd = 0.7%)	Flavanol	LC/UV *
Gallic acid (free)	0.381 ± 0.004 (rsd = 1.1%)	Hydroxybenzoic	LC/UV *
Proanthocyanidins	60 ± 10 (rsd < 2%)	Flavan-3-ol	LC/UV *

Contents in PE60 plum extract were determined either using an Agilent Technologies Model 1200 HPLC System (Wilmington, DE, USA) or a colorimetric method as described in Section 2.2 in the text. Values are mean ± SD of three experiments. * LC/UV = liquid chromatography/ultraviolet light detection

**Table 2 nutrients-11-01077-t002:** Characterization of anti-oxidation properties of PE60.

Assays	Units	Mean ± SD
Total Phenolic Content (TPC)	mg/g	542.44 ± 24.75
Total Flavonoid Content (TFC)	mg/g	520.00 ± 40.10
Anti-oxidant activity (DPPH)	μM Trolox Equivalent/g	3375 ± 90
Oxygen Scavenging Activity (ABTS)	μM Trolox Equivalent/g	4250 ± 250

The anti-oxidation properties of PE60 plum extract were determined using specific assays (TPC: total phenolic content, TFC: total flavonoid content, DPPH: 2,2-diphenyl-1-picrylhydrazyl, ABTS: 2,2’-azino-bis{3-ethylbenzothiazoline-6-sulfonic acid}) as described in Section 2.3 in the text. Values are mean ± SD of three experiments.

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
