# Peer review of "Polyphenol-Enriched Plum Extract Enhances Myotubule Formation and Anabolism while Attenuating Colon Cancer-induced Cellular Damage in C2C12 Cells"

_nutrients, 2019, doi:10.3390/nu11051077_

Reviewer 1 Report

Overall comment: Interesting topic, the manuscript reads very nice and the methods used are very thorough.

Abstract: Please add a line for statistical analysis. 

Introduction: It would be better if authors would explain more about the mechanism of action of plum extract in protecting muscle cells rather than providing examples of animal studies on other functional foods (or their extracts). Each of those anti-oxidant compounds may have their own mechanism of action in vitro and in vivo (considering the differences in their bio-availability). Thus, it will be better if authors focus more on plum extract.

Methods:

-Line 133, what is your rationale for using 0, 50, 100, 150, and 200 g/ml of plum extract?

-Line 119, please add the initial weight of PE60 dissolved in water. 

Results:

-Line 238: Have the authors looked at these results with reference to time? If not, please add an explanation as why time was not considered as a factor in treating the cells. 

Graphs: Please change the format of graphs, if these measurements have been done in different time points and knowing that they are independent measurements (as they are different dose and different batch of cells), it is better to not connect each dot (figure 5 is the correct presentation of the data). 

Discussion: The discussion reads well; however, the authors need to offer a more authentic explanation for their findings.

Author Response

Reviewer #1:

Overall comment: Interesting topic, the manuscript reads very nice and the methods used are very thorough.

Response: All authors thank the reviewer and greatly appreciate his/her comments.

Abstract: Please add a line for statistical analysis. 

Response: A statement for Statistical analysis is now added in the abstract on line 22.

Introduction: It would be better if authors would explain more about the mechanism of action of plum extract in protecting muscle cells rather than providing examples of animal studies on other functional foods (or their extracts). Each of those anti-oxidant compounds may have their own mechanism of action in vitro and in vivo (considering the differences in their bio-availability). Thus, it will be better if authors focus more on plum extract.

Response: Becasue there is almost no data on the effect of plum extract on muscle; we have tried to extrapolate the mechanism of action based on known polyphenols and muscle. In our introduction, we provided a rationale that polyphenols in different fruits and vegetables have been used to modulate muscle metabolism. Much of the possible mechanism of plum extract on skeletal muscle growth is discussed the discussion section.

Methods:

-Line 133, what is your rationale for using 0, 50, 100, 150, and 200 g/ml of plum extract?

Response: Because there is almost no data on the effect of a polyphenol enriched plum extract on muscle cells, we tested various doses to determine if there was dose response. This has been added to lines 137-138 of the revised manuscript.

-Line 119, please add the initial weight of PE60 dissolved in water. 

Response: The initial weight of PE60-dissolved in water (10 mg/ml) is added to line 122 of the revised manuscript.

Results:

-Line 238: Have the authors looked at these results with reference to time? If not, please add an explanation as why time was not considered as a factor in treating the cells

Response: All the experiments were performed at optimized conditions and these conditions are clearly indicated for each experiments. Performing these experiments along time axis is not feasible, since we need to replenish the media in contact with the cells after 24 (along with the plum extract). This will make it difficult to compare and interpret results across a time frame.

Graphs: Please change the format of graphs, if these measurements have been done in different time points and knowing that they are independent measurements (as they are different dose and different batch of cells), it is better to not connect each dot (figure 5 is the correct presentation of the data). 

Response: Although the doses are in continuation from low to high concentration and data is presented in a linear scale, we have revised Figures 1b and 2b as suggested by the reviewer. 

Discussion: The discussion reads well; however, the authors need to offer a more authentic explanation for their findings.

Response: We do not understand what ‘authentic explanation’ implies. Our discussion has cited published references to drive a rational hypothesis about the mechanism of action of the plum extract.  However, we have made changes to the discussion section to allow for a more details explanation, if that would be helpful to the reader. 

Reviewer 2 Report

This manuscript examines the effect of a polyphenol-enriched plum extract on attenuating muscle wasting due to cancer. Specifically, the authors are using a C2C12 in vitro model to assess whether a polyphenol-enriched plum extract enhances anabolism in myoblasts/myocytes and protects differentiated myocytes from the catabolic effects induced by soluble factors released from Colon-26 cancer cells. Understanding how foods or food components might enhance muscle growth and protein synthesis can potentially benefit several populations, especially any populations that experience muscle wasting.

The strengths of this manuscript include:  1) addressing a problem of high importance/need; and 2) providing novel evidence of the muscle anabolic effects of dried plum extract. However, there are some limitations.  First, there is lack of clarity in choice of dried plum extract dosage for each experiment. More explanation of certain doses that were chosen is warranted. For example, most of the results show significant improvements with a 100-200 mg/mL dried plum extract dose, but then only 50 mg/mL of dried plum extract was used in the experiment assessing Colon-26-induced cytotoxicity (Figure 8). Explanation of any discrepancies in dosage use would be helpful. Furthermore, different cells were chosen for the NFkB experiments, instead of assessing the effect of the dried plum extract on NFkB activation in the cells of interest. Therefore, it is unclear what the relevance of this anti-inflammatory experiment is to this model. It is recommended that the anti-inflammatory capacity of the extract be assessed in the C2C12 model.

Introduction:

1.       Page 2, Line 57: Clarify the mouse model—what about TGF in these mice?

2.       Page 2, Line 58: Expand on what is meant by “acting on mitochondria function”

3.       Page 2, Line 63: Consider including an example of a food that is rich in ursolic acid to help the readers out.

4.       Page 2, Line 81-82: Two recent papers demonstrate the effect of dried plum polyphenols on skeletal stem cells—consider citing.

Methods:

1.       Page 3, Line 97: Were these the only polyphenols assessed for? What about other chlorogenic acid isomers reported, and mentioned in the intro, to be in dried plum polyphenol extracts (ie, 4-chlorogenic acid, 5-chlorogenic acid)

2.       Page 3, Line 132: Is “complete” media the same as “differentiation” media? If so, may provide clarity to call this “differentiation” media here.

3.       Page 5, Section 2.9: Why were A549/NFkB-luc cells (from lung epithelium?) chosen for this experiment? Why not assess NFkB activation in your cells of interest (C2C12)? It is unclear what the relevance is, since different cells will respond differently to treatments.

4.       Page 5, Line 205: Provide details on the Colon-26 cells. Where were they obtained from? What kind of colon cell line are they?

Results:

1.       Page 6, Lines 236-237: Consider spelling out the assays to the table can stand alone or conversely at least provide specific details of what section that information can be found.

2.       Page 8, figure 1b: Final asterisk appears to be misplaced.

3.       Page 9, figure 2b: same comment as previous

4.       Page 9, Section 3.3: Why no mention of the statistical significance of the 100 µg/ml dose?

5.       Page 9, Section 3.4: Same comment as previous.

6.       Page 11, Figure 5: Interesting that the control and 50 µg/ml treatment group appear to have the same exact mean and standard error (? Specify in legend how data is being expressed). Just a note to check on this to be sure—may require no change.

7.       Page 11, Figure 6: See previous comments about justification for using this cell line vs assessing the effect of TNF-α on NFκB activation in C2C12 cells. Also, please discuss why the different (much lower) doses were used for this particular experiment.

8.       Page 13, Figure 8: As discussed in the general comments about this manuscript, please describe why 50 µg/ml was chosen as the treatment dose for this experiment when 100-200 µg/ml were more effective in previous experiments.

Discussion

9.       General comment: The flow of the discussion may be easier to follow if it follows the presentation of methods/results. Currently it reads a little choppy.

10.   Page 14, Line 409: Myoblast is misspelled.

11.   Page 14, Line 421: It may not be necessary to start a new paragraph here.

12.   Page 14, Line 421: No “-“ between “which” and “is”

Conclusions:

1.       Page 1015, Line 444: The only anti-inflammatory evidence was not presented from an experiment in which cells were exposed to Colon-26 soluble factors. Additionally, the data was presented in cells that are not myoblasts, raising the question of whether anti-inflammation was demonstrated in this model presented.

Author Response

Please see the MS attached.

Round  2

Reviewer 2 Report

The authors adequately addressed initial concerns, and have enhanced their manuscript. Their findings regarding the effect of a dried plum extract on myotubule formation and anabolism are novel findings and increase our knowledge of the health benefits of dried plums.

 However, their justification for assessing the anti-inflammatory capacity of the dried plum extract should at least be accompanied by acknowledgement of previous literature in which the anti-inflammatory or immune modulating properties of dried plum or its polyphenol extract are demonstrated. There are various papers in which this is the case (some examples from a quick search provided below), and therefore, this finding, which is not demonstrated specifically in their cell model, isn't particularly surprising. Their assay may be unique, but the anti-inflammatory and antioxidant capacity of dried plums has been studied and demonstrated in other models, especially bone models. This previous evidence and why they chose to assess using their chosen assay might be discussed in more detail in the Discussion section paragraph including lines 415-422.

Bu, So Young, et al. "Dried plum polyphenols inhibit osteoclastogenesis by downregulating NFATc1 and inflammatory mediators." Calcified tissue international 82.6 (2008): 475-488.

Hooshmand, Shirin, et al. "Evidence for anti-inflammatory and antioxidative properties of dried plum polyphenols in macrophage RAW 264.7 cells." Food & function 6.5 (2015): 1719-1725.

Rendina, Elizabeth, et al. "Dietary supplementation with dried plum prevents ovariectomy-induced bone loss while modulating the immune response in C57BL/6J mice." The Journal of nutritional biochemistry 23.1 (2012): 60-68.

Author Response

Response:  This is an excellent suggestion by reviewer and we thank him/her for raising this issue.  We have performed extensive search on “Web of Science” and added the following paragraph on lines 415 -419. 

“Studies have demonstrated anti-inflammatory effect of dried-plum or plum juice in several cellular system including lipopolysaccharide-induced macrophages [64, 65], splenocytes from ovariectomized mice [66], colorectal cells in azoxymethane-treated rats [67], heart tissues in obese rats [68] and joints of TNF-over expressing mice [69]. The antioxidation activities of plum appeared to be mediated through the inhibition of NFkB activation [67, 68, 69].”

64. Bu, S.Y.; Lerner, M.; Stoecker, B.J.; Boldrin, E.; Brackett, D.J.; Lucas, E.A.; Smith, B.J. Dried plum polyphenols inhibit osteoclastogenesis by downregulating NFATc1 and inflammatory mediators. Calcif Tissue Int. 2008, 82, 475-88.

65. Hooshmand, S.; Kumar, A.; Zhang, J.Y.; Johnson, S.A.; Chai, S.C.; Bahram H. Arjmandi. B.H. Evidence for anti-inflammatory and antioxidative properties of dried plum polyphenols in macrophage RAW 264.7 cells. Food & Function 2015, 6, 1719-1725.

66. Rendina, E.;  Lima, Y.F.; Marlowb, D.; Wanga, Y.; Clarkea, S.L.; Kuvibidilaa, S.; Lucasa, E.A.;  Brenda J. Smith, B.J. Dietary supplementation with dried plum prevents ovariectomy-induced bone loss while modulating the immune response in C57BL/6J mice. The J. Nutr. Biochem. 2012, 23, 60-68.

67. Banerjee, N.; Kim, H.; Talcott, S.T.; Turner, N.D.; Byrne, D.H.; Mertens-Talcott, S.U. Plum polyphenols inhibit colorectal aberrant crypt foci formation in rats: potential role of the miR-143/protein kinase B/mammalian target of rapamycin axis. Nutr. Res. 2016, 36, 1105-1113.

68. Noratto, G.; Martino, H.S.D.; Simbo, S.; Byrne, D.; Mertens-Talcott, S.U. Consumption of polyphenol-rich peach and plum juice prevents risk factors for obesity-related metabolic disorders and cardiovascular disease in Zucker rats. The J. Nutr. Biochem. 2015, 26, 633-641     

69. Mirza, F.; Lorenzo, J.; Drissi, H.; Lee, F.Y.; Soung, D.Y. Dried plum alleviates symptoms of inflammatory arthritis in TNF transgenic mice. The J. Nutr. Biochem  2018, 52, 54-61.